# Plasma membrane-derived extracellular microvesicles mediate non-canonical intercellular NOTCH signaling

Qiyu Wang[1] & Quan Lu[1,2]

ARMMs (arrestin domain-containing protein 1 (ARRDC1)-mediated microvesicles) are extracellular vesicles that bud directly at the plasma membrane; however, little is known about the molecular composition and physiological function of these vesicles. Here we report that ARMMs contain active NOTCH receptors and mediate a non-canonical intercellular NOTCH signaling. We identify over 100 proteins that are significantly enriched in ARMMs, including ARRDC1, TSG101 and multiple ESCRT complex proteins. About a third of ARMMs-enriched proteins are plasma membrane proteins, including the NOTCH2 receptor. The incorporation of NOTCH2 into ARMMs is facilitated by the ITCH E3 ligase and the metalloprotease ADAM10, both of which are also secreted into ARMMs. NOTCH2 in ARMMs can be delivered into recipient cells, and upon activation by γ-secretase cleavage, induces NOTCH-specific gene expression. Together, our findings reveal a role for ARMMs in a novel NOTCH signaling pathway that acts in distance and is independent of direct cell–cell contact.

[1] Program in Molecular and Integrative Physiological Sciences, Department of Environment Health, Harvard T.H. Chan School of Public Health, 665 Huntington Ave., Boston, MA 02115, USA. [2] Department of Genetics & Complex Diseases, Harvard T.H. Chan School of Public Health, 665 Huntington Ave., Boston, MA 02115, USA. Correspondence and requests for materials should be addressed to Q.L. (email: qlu@hsph.harvard.edu)

It is now widely appreciated that cells from multicellular organisms are capable of secreting into extracellular milieu membrane-encapsulated small vesicles[1, 2]. These extracellular vesicles (EVs) carry a myriad of molecules from the host cells, including proteins and RNAs[3–5]. As such, the EVs have the capacity to transfer these molecules to initiate signaling in recipient cells or tissues. Indeed, studies over the past decade have implicated EVs as a new way of cell–cell communication in a variety of physiological and disease settings, including cancer and immune response[6–8].

EVs are heterogeneous, comprising of vesicles with distinct mechanisms of biogenesis. Most studies have focused on exosomes, which originate in the late endosomes as multivesicular bodies (MVBs)[9]. Through a process known as exocytosis, MVBs can traffic to the cell surface to fuse with the plasma membrane, resulting in the secretion of MVBs as exosomes[10]. In contrast, some EVs can be directly generated at the plasma membrane[11]. We have previously identified a novel type of EVs known as ARMMs that originate at the plasma membrane[12]. The budding of ARMMs requires arrestin-domain containing protein 1 (ARRDC1), which is localized to the cytosolic side of the plasma membrane and recruits the ECSCRT I complex protein TSG101 to initiate the budding of the vesicles[12]. Although the physiological function of ARMMs remains unknown, the biogenesis at the plasma membrane suggests that ARMMs may be uniquely situated to carry plasma membrane-associated molecules such as functional receptors to mediate intercellular signaling.

NOTCH receptors are highly conserved plasma membrane proteins that mediate a wide range of critical physiological roles such as embryonic development, tissue homeostasis, immunity and stem cell function[13, 14]. Canonical NOTCH signaling requires direct contact between neighboring cells[15]. The core NOTCH pathway is initiated through the binding of the receptors on one cell to the NOTCH ligands located on the surface of another cell[16]. This intercellular receptor–ligand interaction triggers proteolytic cleavage of NOTCH, resulting in the release of the NOTCH intracellular domain (NICD). Translocation of NICD into the nucleus activates the transcription of multiple NOTCH target genes, such as HES1 and HES5[17]. In addition to this classical pathway, NOTCH can also be activated intracellularly in the endosomal compartments thorough a non-canonical pathway that is ligand-independent[18], as disruption of the endosomal-sorting machinery triggers the NOTCH signaling[19–21].

Despite the existence of multiple pathways of NOTCH activation, it is not known whether NOTCH receptors can signal beyond the neighboring cells in contact, and what mechanisms may exist to carry out such non-canonical NOTCH signaling. Here we provide evidence that NOTCH receptors are specifically recruited into ARMMs for extracellular release. More importantly we show that NOTCH contained in ARMMs can be transferred to recipient cells to mediate specific NOTCH signaling. Our study thus identifies an important physiological role for ARMMs and reveals a novel NOTCH receptor signaling pathway that is mediated by EVs and is independent of direct cell–cell contact.

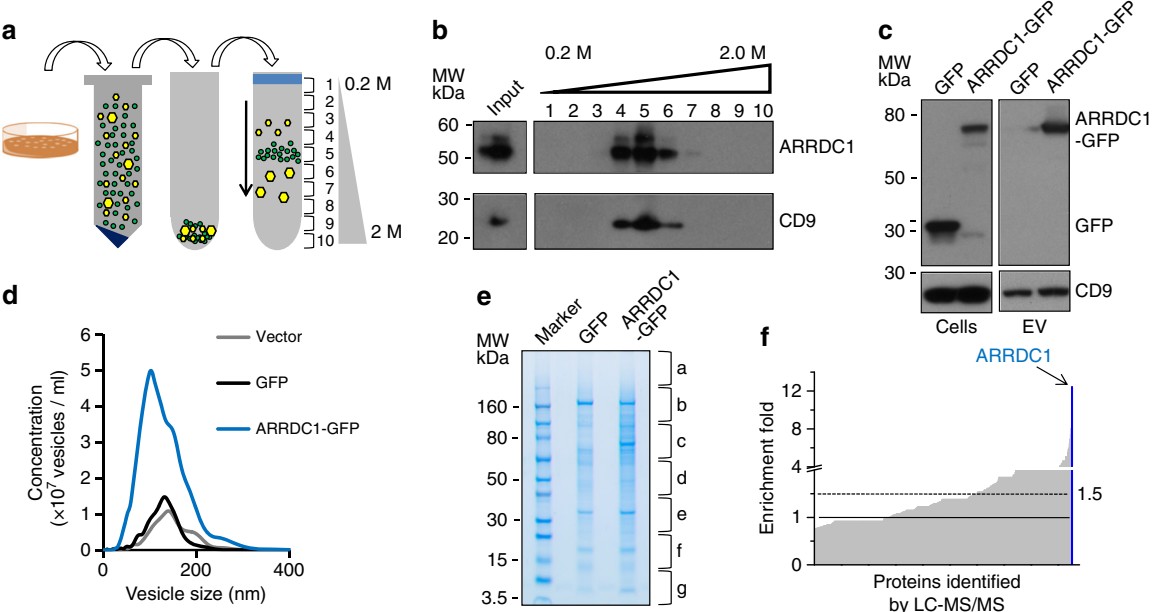

**Fig. 1** Identification of protein components in ARMMs. **a** Schematic procedure used to isolate ARMMs. Conditioned medium from control or transfected HEK293T cells was pre-cleared and centrifuged. Vesicles were first pelleted and then put onto a sucrose gradient for ultracentrifugation. Ten fractions were collected. **b** Western blotting of fractionated samples. Each of the ten fractions from sucrose gradient ultracentrifugation was spun down by ultracentrifugation. Pellets were then used for anti-ARRDC1 and anti-CD9 western blotting. **c** Expression and budding of ARRDC1-GFP fusion protein. HEK293T cells were transfected with GFP or ARRDC1-GFP expression conctructs. Extracellular vesicles (EV) were pelleted by ultracentrifugation. Both cell lysates and vesicle pellets were used for anti-GFP and anti-CD9 western blotting. **d** NanoSight analysis of EVs from HEK293T cells transfected with either vector control, GFP or ARRDC1-GFP constructs. **e** SDS-PAGE of proteins from EVs. Vesicles were isolated from GFP or ARRDC1-GFP-transfected HEK293T cells using the sucrose gradient method. Peak ARMMs fractions were collected and resolved on SDS-PAGE. Protein bands were visualized by Coomassie blue staining. Eight gel slices from each lane (GFP or ARRDC1-GFP) were obtained and used for mass-spectrometry. **f** Graph showing protein enrichment in ARMMs. LC-MS/MS was used to identify proteins in EVs from GFP or ARRDC1-GFP-expressing HEK293T cells. Peptide numbers for each of the proteins were used to derive the enrichment fold (ARRDC1-GFP EV vs. GFP EV). Proteins were ranked by enrichment fold (low to high on the X-axis). ARRDC1 is the most enriched protein as indicated

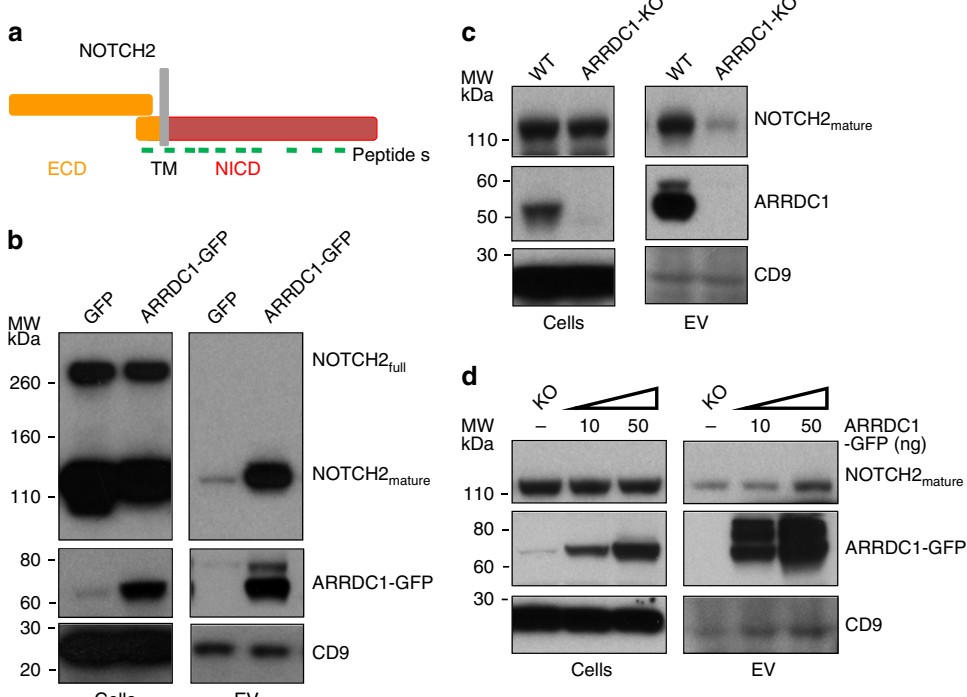

**Fig. 2** NOTCH2 receptor is secreted into ARMMs. **a** Schematic drawing of domains in NOTCH2 receptor. Relative positions of peptides identified by mass spectrometry were indicated. ECD: extracellular domain; TM: transmembrane domain; NICD: NOTCH intracellular domain. **b** Western blotting of NOTCH2 protein in ARMMs. HEK293T cells were transfected with GFP or ARRDC1-GFP. Extracellular vesicles were pelleted by ultracentrifugation. Both cell lysates and extracellular vesicle (*EV*) pellets were used for anti-NOTCH2, anti-ARRDC1 and anti-CD9 western blotting. **c** Effect of ARRDC1 knockout on NOTCH release into ARMMs. CRISPR was used to knock out the ARRDC1 gene in HEK293T cells. *EVs* from conditioned culture media of ARRDC1-KO and wild-type HEK293T cells were pelleted and subjected to western blotting along with corresponding cell lysates. **d** Re-expression of ARRDC1 restored NOTCH2 release into ARMMs in ARRDC1-KO cells. ARRDC1-KO HEK293T cells were transfected with the indicated amount of ARRDC1-GFP construct. Two days after transfection, *EVs* were pelleted from the conditioned media and subjected to western blotting along with the cell lysates

| Table 1 Enrichment of NOTCH pathway components in ARMMs | | | |
| --- | --- | --- | --- |
| **Protein ID** | **GFP** | **ARRDC1-GFP** | **Ratio** |
| ARRDC1 | 61 | 821 | 12.42 |
| ITCH | 7 | 61 | 8.04 |
| NOTCH2 | 5 | 18 | 3.32 |
| NOTCH1 | 0 | 4 | – |
| ADAM10 | 17 | 35 | 1.90 |
| CD9 | 12 | 13 | 1.00 |
| CD81 | 55 | 59 | 0.99 |

Peptide numbers for each protein from EVs were shown. The ratio represents peptide number from ARRDC1-GFP EVs (normalized against CD9) divided by peptide numbers from control GFP EVs. Exosome-associated proteins CD9 and CD81 were included as controls

## Results

**Identification of protein components of ARMMs.** In order to understand the physiological role of ARMMs in cell signaling, we set out to identify the molecular components, in particular proteins, in the vesicles. Because EVs secreted by cells are heterogeneous, we first attempted to separate ARMMs from other EVs using sucrose gradient ultracentrifugation. Pre-cleared EV pellet from HEK293T cells was separated by ultracentrifugation on a sucrose gradient (0.2 M to 2 M sucrose) into 10 fractions (Fig. 1a). EVs in each of the 10 fractions were then pelleted by ultracentrifugation again and analyzed by western blotting. As shown in Fig. 1b, ARMMs as indicated by ARRDC1 immuno-blotting were detected in fractions 4–6 with a peak in fraction 5. However,

consistent with the finding that the size of ARMMs is similar as that of exosomes[12], the ARRDC1-positive fractions also contained CD9, a tetraspanin protein commonly found in EVs and sometimes used as a marker for exosomes[22]. This indicated that, although sucrose gradient ultracentrifugation was able to fractionate ARMMs, the method did not separate ARMMs from exosomes.

The impurity of the sucrose fractions did not allow us to distinguish and identify specific components in ARMMs. To circumvent this problem we took advantage of the fact that the expression of ARRDC1 alone is sufficient to drive the budding and production of ARMMs[12]. We reasoned that we may be able to compare the EVs from control and ARMMs-overproduction cells to delineate the specific components of ARMMs. As expected, the ARRDC1-GFP fusion protein, when expressed into HEK293T cells, is able to bud out of cells into EVs (Fig. 1c). Moreover, ARRDC1-GFP-transfected cells produced significantly more (~3-fold) EVs compared to control or GFP-transfected cells (Fig. 1d), indicating that ARRDC1 overexpression drives the production of new ARMMs. We then fractionated the EVs from both GFP and ARRDC1-GFP-expressing HEK293T cells using the sucrose gradient ultracentrifugation method. Peak fractions containing ARMMs were collected, and proteins contained in the EVs were resolved on sodium dodecyl sulfate polyacrylamide gel electrophoresis (SDS-PAGE) gel (Fig. 1e) and subjected to mass spectrometry (LC-MS/MS).

We reasoned that proteins specifically associated with ARMMs would be enriched in EVs from ARRDC1-GFP-expressing HEK293T cells. We compared the LC-MS/MS results of the

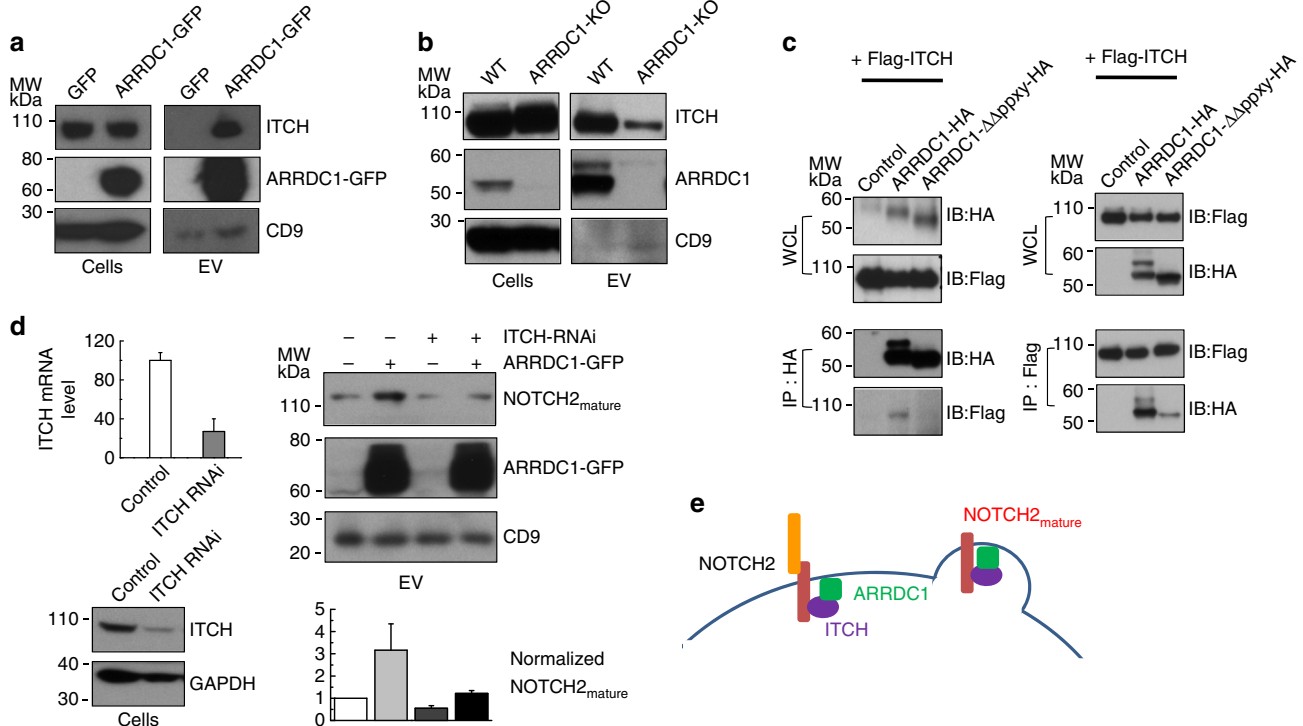

**Fig. 3** ITCH interacts with ARRDC1 and mediates NOTCH2 incorporation into ARMMs. **a** ARMMs contain ITCH. HEK293T cells were transfected with GFP or ARRDC1-GFP. EVs were pelleted by ultracentrifugation. Both cell lysates and EVs were used for anti-ITCH, anti-ARRDC1 and anti-CD9 western blotting. **b** Effect of ARRDC1 knockout on ITCH release into EVs. EVs from conditioned culture media of ARRDC1-KO and wild-type HEK293T cells were pelleted and subjected to anti-ITCH, anti-ARRDC1 and anti-CD9. **c** Co-immunoprecipitation (*IP*) showing the interaction between ARRDC1 and ITCH. HEK293T cells were co-transfected with Flag-tagged ITCH and one of the following: control vector, HA-tagged-ARRDC1 or ARRDC1-delPPXY (deletion of two PPXY motifs). Two days after transfection, cell lysates were collected and incubated with anti-HA agarose beads for immunoprecipitation. IP-ed complexes were subjected to anti-HA or anti-Flag western blotting. **d** Effect of ITCH knockdown on NOTCH2 release into ARMMs. HEK293T cells were co-transfected with ITCH siRNA (or scrambled control) and ARRDC1-GFP (or control GFP). Three days after transfection, EVs from conditioned culture media were pelleted and subjected to western blotting. The amount of NOTCH2 in EV was quantified. Data represent an average of three independent repeats. **e** Model depicting the role of ITCH on NOTCH2 release into ARMMs

EVs from GFP and ARRDC1-GFP cells. Based on normalized peptide numbers, we identified a total of 177 proteins whose presence in ARRDC1-GFP EVs is increased (>1.5-fold) (Fig. 1f and Supplementary Data 1). In addition, 65 proteins (with at least 5 peptides) were present only in ARRDC1-GFP EV but not detected in the control EVs (Supplementary Data 2). As expected, the most enriched protein is ARRDC1 (Table 1). Another group of enriched proteins include TSG101 and other components of the ESCRT complexes (FAM125A, CHMP3, CHMP6, CHMP1B, VPS37A,B,C,D, VPS25, VPS28 and VPS36) (Supplementary Data 3), which normally mediate the endosomal sorting of ubiquitinated membrane proteins[23]. Several NEDD4 E3 ligases (WWP1, WWP2 and ITCH) are also enriched in ARMMs. Other enriched proteins include several proteasome components and members of the integrin family. Finally, we note that the exosome makers (CD9 and CD81) did not change significantly between control and ARRDC1 EVs (Table 1).

**NOTCH receptors are released into ARMMs.** Strikingly, about a third of the ARMMs-enriched proteins (54 out of 177) are plasma membrane proteins (Supplementary Data 4). This is consistent with the notion that ARMMs, unlike exosomes, are generated at and bud directly from the cell surface. Among the enriched plasma membrane proteins are NOTCH receptors (NOTCH2 and NOTCH1) (Table 1). Interestingly, ITCH, the E3 ligase that interacts with and ubiquitinates NOTCH proteins[24, 25], is among

the most enriched proteins in ARMMs (Table 1). In addition, ADAM10—the metalloprotease that cleaves and activates NOTCH receptors[26, 27]—is also enriched in ARMMs (Table 1). Together, these results indicated that ARMMs contain multiple components of the NOTCH signaling pathway. We note that NOTCH receptors had not been found in exosomes in the Exo-Carta database[22], suggesting that the receptors are ARMMs specific.

Because we identified more peptides from NOTCH2 than from NOTCH1, we focused our studies on the NOTCH2 receptor. Mass spectrometry identified a total of 18 peptides for NOTCH2 in ARMMs. Interestingly, most of the NOTCH2 peptides were located in the C-terminus of the receptor protein (Fig. 2a and Supplementary Fig. 1). It is well known that NOTCH receptors, once translated, are proteolytically cleaved by furin to generate mature receptor proteins[28], which are heterodimers of the two cleaved products held together non-covalently[29]. Thus our mass spectrometry result suggests that ARMMs contain the transmembrane (TM) domain and the NICD but not the extracellular component of the NOTCH heterodimer. This was confirmed by western blotting: whereas both full-length and short forms of NOTCH2 were present in the cells, only the short (mature) form (TM plus NICD) was detected in ARMMs (Fig. 2b). Incorporation of NOTCH short form into EVs was also shown in several other cell lines such as HCC1419, A549 and MCF-7 cells (Supplementary Fig. 2a).

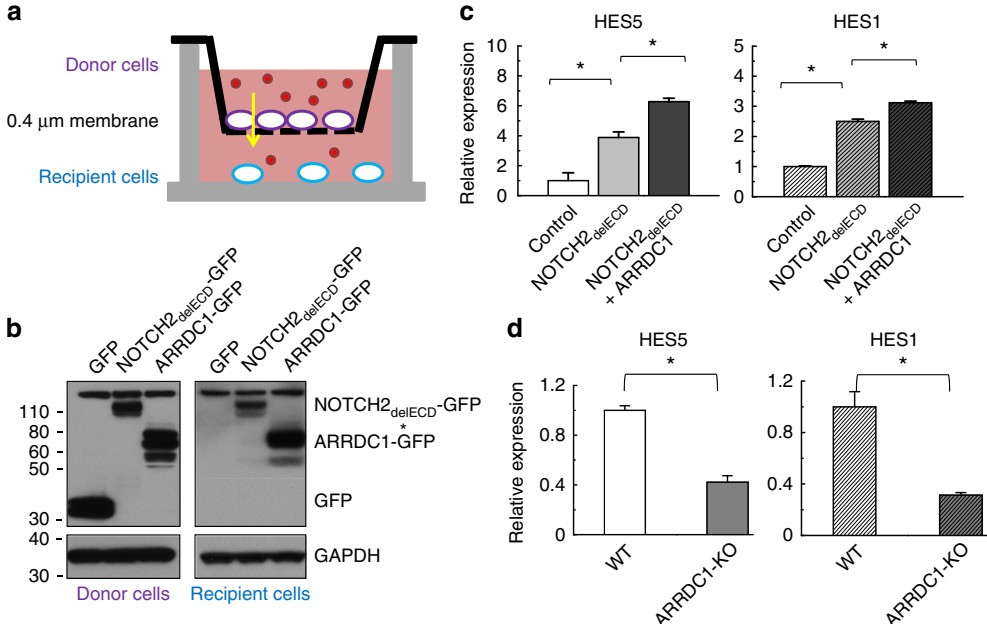

**Fig. 4** ARMMs deliver functional NOTCH2 to recipient cells. **a** Schematic drawing of the transwell assay in which cultured recipient and donor cells were separated by a 0.4 μm membrane. **b** Western blotting of NOTCH2 (GFP-tagged) in donor and recipient cells. HEK293T cells were transfected with GFP, NOTCH2delECD-GFP (NOTCH2 used here does not contain the extracellular domain) and ARRDC1-GFP. Twenty-four hours later, the transfected cells were seeded as donor cells in transwells and co-cultured with recipient HEK293T cells. Recipient cells were washed with PBS and lysed for anti-GFP western blotting analysis. GAPDH western blotting was done to ensure equal loading. Asterisk indicates a non-specific protein band. **c** Expression of NOTCH target genes in recipient cells. Donor HEK293T cells were transfected with GFP, NOTCH2delECD-GFP or NOTCH2delECD-GFP plus HA-ARRDC1, and co-cultured with recipient HEK293T cells in transwells for 48 h. RNAs were then extracted from the recipient cells and used for qRT-PCR to measure the expression of HES1 and HES5 genes. **d** qRT-PCR data showing HES1/HES5 mRNA levels in recipient cells that received ARMMs from control or ARRDC1-KO HEK293T cells. Data represent an average of three independent repeats for each condition. *Error bars* indicate standard deviation (s.d.). *$p < 0.05$

To further confirm that NOTCH2 is specifically released into ARMMs, we investigated whether the extracellular release of NOTCH2 requires ARRDC1. Using the CRISPR gene editing technique[30], we created HEK293T cells with complete *ARRDC1* knockout. As shown in Fig. 2c, the amount of NOTCH2 released into the EVs dramatically decreased in ARRDC1-knockout cells, while the NOTCH2 receptor level remained unchanged in the knockout cells. To further ascertain the role of ARRDC1 in NOTCH2 release into EVs, we performed reconstitution experiment in the ARRDC1-knockout cells. As shown in Fig. 2d, re-expression of ARRDC1 in ARRDC1-knockout cells rescued the NOTCH2 release in EVs. Consistent with these findings in HEK293T cells, siRNA-mediated knockdown of ARRDC1 in MCF-7 cells also significantly reduced the amount of NOTCH2 short form in the EVs (Supplementary Fig. 2b). Together these experiments demonstrated that the release of NOTCH2 into EVs is dependent on ARRDC1.

**NOTCH release into ARMMs requires the ITCH E3 ligase.** ITCH, a member of the NEDD4 E3 ligase family that interacts with and ubiquitnates NOTCH receptors[24, 31], is a top enriched protein in ARMMs (Fig. 2a). Western blotting confirmed the presence of ITCH in ARMMs (Fig. 3a). Similar to the release of NOTCH2, ITCH release in EVs also requires ARRDC1, as the knockout of ARRDC1 significantly reduced the amount of ITCH in EVs (Fig. 3b). Consistent with a previous study[31], co-immunoprecipitation (IP) assay in both directions (i.e., ITCH and ARRDC1 pull downs by anti-Flag antibody and anti-HA antibody, respectively) showed that ARRDC1 interacts with ITCH and that this interaction is abolished by the deletion of the PPXY motifs in ARRDC1 (Fig. 3c).

Given that ITCH interacts with both ARRDC1 and NOTCH, we hypothesize that ITCH may mediate the recruitment of NOTCH2 into ARMMs. To test this, we use siRNAs to knockdown ITCH and then examined the NOTCH2 release into ARMMs. As shown in Fig. 3d (*left panel*), siRNAs efficiently knocked down ITCH mRNA and protein by more than 60%. ITCH knockdown significantly reduced the amount of NOTCH2 in ARMMs both at the basal level and under the condition of ARRDC1 overexpression (Fig. 3d, *right panel*). This result indicates an important role for ITCH in the release of NOTCH2 into ARMMs (Fig. 3e).

**Functional transfer of NOTCH2 into recipient cells via ARMMs.** When incubated with recipient cells, EVs are taken up efficiently and fairly quickly—sometimes within 30 min[32, 33]. We performed confocal imaging and found that the uptake of ARMMs by the recipient HEK293T cells also occurs quickly—within 2 h of incubation (Supplementary Fig. 3). To test whether the NOTCH2 receptor secreted into ARMMs can be transferred into recipient cells, we employed a transwell-based assay (Fig. 4a), in which donor cells producing ARMMs were cultured separately from the recipient cells. The 0.4 μM transwell membrane separating the donor and recipient cells allows small EVs such as ARMMs to pass through. We then expressed either control GFP, NOTCH2delECD-GFP (NOTCH2 used here lacks the extracellular domain (ECD)), or ARRDC1-GFP in the donor cells. After co-culturing of the donor cells with recipient HEK293T cells that do not express any of the GFP proteins, we used western blotting to detect the presence of GFP and the GFP-fusion proteins in the recipient cells. As expected, ARRDC1-GFP, which by itself can generate ARMMs, was robustly detected in the recipient cells,

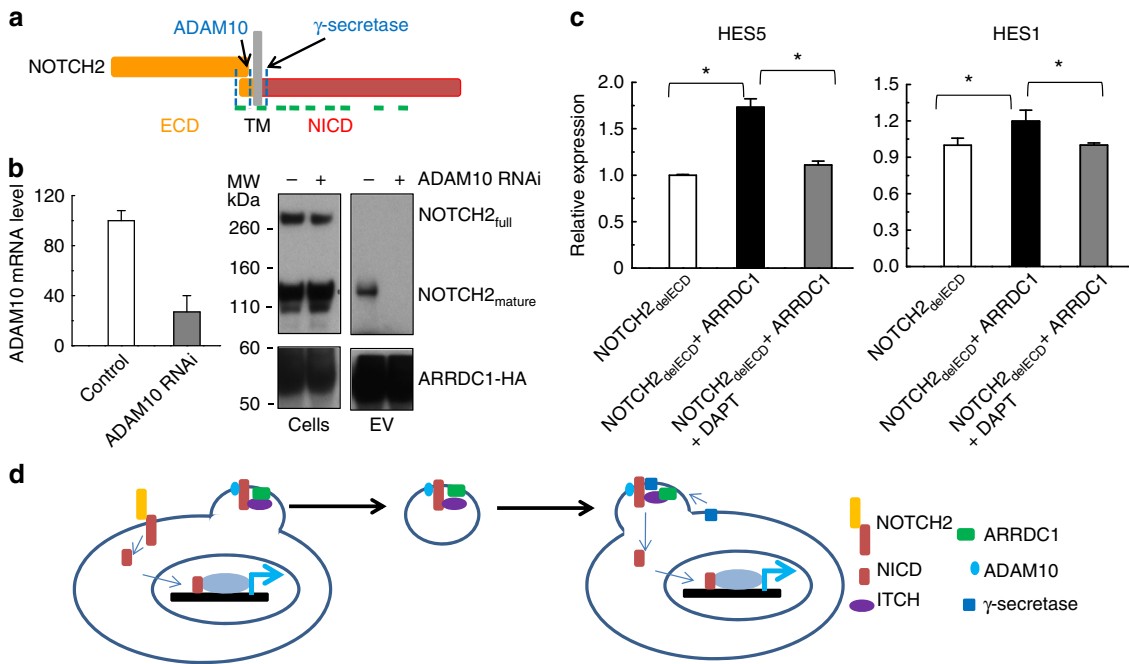

**Fig. 5** Roles of ADAM10 and γ-secretase in NOTCH2 release and activation. **a** Schematic drawing of NOTCH2 receptor with indicated cleavage sites for ADAM10 and γ-secretase. **b** Effect of ADAM10 knockdown on NOTCH2 release into ARMMs. *Left panel*: qRT-PCR data showing siRNA-mediated knockdown of ADAM10 in HEK293T cells. *Right panel*: Western blot analysis. HEK293T cells were transfected with scrambled control or ADMA10 siRNAs. EVs and cell lysates were collected and subjected to western blotting. **c** Effect of γ-secretase inhibitor on NOTCH activation in recipient cells. HEK293T donor cells were transfected with NOTCH2delECD-GFP alone or together with HA-ARRDC1. Donor cells were then co-cultured with recipient HEK293T cells in the absence or presence of 100 nM γ-secretase inhibitor DAPT. qRT-PCR data showed the expression of HES genes in recipient cells. Data represent an average of three independent repeats for each condition. *Error bars* indicate standard deviation (s.d.). *$p < 0.05$. **d** Proposed model for ARMMs-mediated non-canonical NOTCH2 signaling

whereas the GFP protein alone was not (Fig. 4b). Importantly, NOTCH2delECD-GFP was detected in the recipient cells (Fig. 4b), indicating the NOTCH2 protein was transferred to the recipient cells.

We next determined whether the NOTCH2 receptor transferred into recipient cells is capable of initiating signaling. To do this, we measured the expression of NOTCH2 target genes (HES1 and HES5) in recipient cells in the transwell experiment. In recipient cells co-cultured with donor cells expressing NOTCH2, we found that both HES1 and HES5 were significantly increased, comparing to that in the control recipient cells co-cultured with GFP-expressing donor cells (Fig. 4c). Furthermore, when we co-expressed both NOTCH2 and ARRDC1 together in donor cells, we detected even higher up-regulation of the HES genes in the recipient cells (Fig. 4c). To further ascertain that the NOTCH2 target gene expression in the recipient cells is dependent on ARMMs, we repeated the assay using ARRDC1-knockout cells as donor cells. We found that the expression of NOTCH2 in ARRDC1-knockout donor cells failed to induce the expression of HES1 or HES5 in the recipient cells (Fig. 4d). Together, these experiments demonstrate that NOTCH2 receptor can be transferred to and initiate signaling in recipient cells and that such transfer is dependent on ARMMs.

**Activation of ARMMs-transferred NOTCH2 requires γ-secretase.** Canonical NOTCH activation requires sequential proteolytic cleavages of the receptor by α-secretase ADAM10 and by γ-secretase[15] (Fig. 5a). These cleavages ultimately lead to the release of the transcriptionally active NICD. Although NOTCH2 in ARMMs lacks the ECD, it still contains a small extracellular segment as well as the TM domain. Thus the NOTCH2 receptor

in ARMMs likely needs to be cleaved by both ADAM10 and the γ-secretase into NICD to be activated. Interestingly, ADAM10 is one of the proteins detected and moderately enriched in ARMMs (Fig. 2a). Since ADAM10 is already in the ARMMs, it is unlikely that NOTCH needs to be cleaved by ADAM10 in the recipient cells. We thus decided to test whether ADAM10 from the donor cells is required for NOTCH2 release into ARMMs. We used RNAi to knock down ADAM10 and determined the effect on NOTCH2 release. When ADAM10 was knocked down by over 60%, NOTCH2 was no longer able to get into the ARMMs (Fig. 5b).

We next determined whether γ-secretase is required for the activation of transferred NOTCH receptor in the recipient cells. To do this, we used the specific γ-secretase inhibitor DAPT[34, 35]. NOTCH-containing ARMMs from production donor cells were used to incubate with recipient cells in the presence or absence of DAPT. As expected, in the absence of the inhibitor, incubation with NOTCH-ARMMs led to increased expression of both HES1 and HES5 genes. However, in the presence of DAPT, the increase in the NOTCH target gene expression was abolished (Fig. 5c). This result suggests that NOTCH2 transferred by ARMMs into recipient cells is likely cleaved by the γ-secretase into the NICD to activate the target gene expression.

**Discussion**

ARMMs are novel EVs that bud directly from plasma membrane[12, 36]. Unlike the exosomes, little is known about the molecular composition and function of ARMMs. In this study we identified the major protein components of ARMMs and demonstrated that NOTCH receptors are recruited into ARMMs and can be transferred to recipient cells to mediate specific

NOTCH signaling. Our study revealed the general protein makeup of ARMMs and identified a novel role for ARMMs in non-canonical NOTCH signaling.

ARMMs bud directly at the plasma membrane and are distinct from exosomes—the classical EVs. Supporting the non-exosomal nature of ARMMs, general exosomal markers such as CD9 and CD81 were not enriched in ARMMs; neither did we detect significant amount of other proteins that are frequently found in exosomes. Consistent with the central and essential role of ARRDC1 in ARMMs budding[12, 36], we identified ARRDC1 as the most enriched protein in the vesicles. Another group of enriched proteins include TSG101 and multiple components of the ESCRT complexes (CHMP and VPS proteins). This is consistent with our previous finding that TSG101 is recruited by ARRDC1 and is required for ARMMs budding[12], and further suggests that other specific ESCRT components may be involved in the budding of ARMMs. We have previously shown that ARRDC1 interacts with NEDD4 family E3 ligases and that ARMMs production is enhanced by these interactions[12]. In agreement with this, we identified several NEDD4 E3 ligases (WWP1, WWP2 and ITCH) as enriched in ARMMs. Other enriched proteins include proteasome components and several members of the integrin family. Future studies are needed to determine whether these proteins modulate ARMMs budding or are just bystanders of budding.

Consistent with their biogenesis, ARMMs contain a myriad of plasma membrane proteins, including the NOTCH receptors. We showed that ARMMs can deliver NOTCH receptors to induce specific NOTCH signaling in recipient cells. Canonical NOTCH signaling is initiated when the ECD of the NOTCH receptor on one cell is engaged with its specific ligand sitting on the cell surface of another cell[15]. As such, the NOTCH activation is usually dependent on the direct contact between the two neighboring cells. Contrary to this canonical view of NOTCH signaling, our results support a model in which ARMMs enable the NOTCH receptor to signal beyond neighboring cells in contact (Fig. 5d). In this model, NOTCH receptor, through interaction with ITCH, which also interacts with ARRDC1, is recruited into ARMMs for extracellular release. The recruitment of NOTCH into the ARMMS also requires ADAM10, which cleaves NOTCH at the S2 site. NOTCH contained in ARMMs can be brought to distant sites to engage with recipient cells. While it remains to be determined whether ARMMs deliver NOTCH into recipient cells via endocytosis or direct fusion, it is clear that NOTCH is further cleaved by γ-secretase into the active NICD form, which translocates into nucleus to induce target gene expression. This ARMMs-mediated process bypasses the canonical requirement of direct cell–cell contact, thus expanding the range of NOTCH signaling. Interestingly, another study showed that the NOTCH ligand Delta-like 4 carried onto exosomes is able to modulate NOTCH signaling[37], supporting the concept of NOTCH signaling at a distance. Future in vivo studies will further establish the physiological role of EVs including ARMMs in non-canonical NOTCH signaling.

ARMMs likely possess the capability to deliver other signaling molecules. Indeed, our mass spectrometry results have identified many other plasma membrane proteins such as integrins and growth factor receptors in ARMMs. Although some of these proteins have been identified in exosomes, their enrichment in ARMMs suggests that they may preferably bud into ARMMs. Future work should elucidate the recruitment mechanism and the corresponding signaling capability. In addition to proteins, RNA molecules, which have been identified in EVs, are also likely contained in ARMMs. Identification and further characterization of these molecules in ARMMs will expand the repertoire of signaling capacity of ARMMs. Together these studies will illuminate the physiological function of ARMMs and establish the vesicles as a new paradigm in cell–cell communication.

## Methods

**Cell culture, transfections and chemical inhibitors**. HEK293T, HCC1419, A549 and MCF-7 cells were obtained from the American Type Culture Collection (ATCC). HCC1419 cells were maintained and cultured in RPMI1640 supplemented with 10% fetal bovine serum (FBS) (Gibco). All the other cells were maintained and cultured in Dulbecco's Modified Eagle's medium supplemented with 10% FBS (Gibco). Cell cultures are routinely checked for mycoplasma contamination. In experiments where ARMMs were to be collected, FBS was pre-cleared of EVs by ultracentrifugation (at 188,000×$g$ for 4 h). SMARTpool siRNAs for ITCH and ADAM10 were purchased from Dharmacon (GE Healthcare). Transfections of siRNAs were done using the Dharmafect reagent (GE Healthcare). DNA transfections were done using the Turbofect reagent (Thermo Fisher Scientific). The γ-secretase inhibitor DAPT and the dynamin inhibitor Dynasore were purchased from Sigma and Tocris, respectively.

**Plasmid constructs**. ARRDC1-GFP, ITCH (AIP4) and ARRDC1-delPPXY were described previously[12]. NOTCH2$_{delECD}$-GFP was constructed by cloning the C-terminal NOTCH2 (amino acids 1678–2471 of full-length NOTCH2) along with the single peptide (amino acid 1–26) into the NheI and AgeI sites of the pEGFP-N1 vector (Clontech).

**Antibodies, western blotting and immuno-precipitation**. Primary antibodies used in the study include rabbit polyclonal ARRDC1 antibody as described previously[12], mouse monoclonal CD9 antibody (Santa Cruz, catalogue #sc-13118, at 1:1000 dilution), rabbit monoclonal GFP antibody (Cell Signaling, catalogue #2956S, at 1:1000 dilution) and rabbit monoclonal NOTCH2 and ITCH antibodies (Cell Signaling, catalogue #4530S, 12117S, both at 1:2000 dilution). Second antibodies were anti-rabbit IgG HRP-linked antibody (Cell Signaling, catalogue #7074S, at 1:5000 dilution) and anti-mouse IgG HRP-linked antibody (Cell Signaling, catalogue #7076S, at 1:500 dilution). Western blotting and co-immuno-precipitation were done as previously described[12]. Briefly, HEK293T cells were co-transfected with Flag-tagged ITCH and one of the following: control vector, HA-tagged-ARRDC1 or ARRDC1-delPPXY (deletion of two PPXY motifs). Two days after transfection, cell lysates were collected and incubated with anti-HA agarose beads for immunoprecipitation. IP-ed complexes were subjected to anti-HA or anti-Flag western blotting. Uncropped blots are shown in Supplementary Fig. 4.

**RNA extraction and quantitative real-time PCR**. RNA extractions were done using the Trizol reagent (Thermo Fisher Scientific). Reverse transcription was done using the SuperScript III Reverse Transcriptase kit (Thermo Fisher Scientific). Primer sequences for the qPCR of HES1 and HES5 are Hes1-F: TCAACACGAC ACCGGATAAA, Hes1-R: TCAGCTGGCTCAGACTTTCA, Hes5-F: CCGGTGG TGGAGAAGATG and Hes5-R: TAGTCCTGGTGCAGGCTCTT. qPCR was performed on the StepOnePlus Real-Time PCR System (Life) with SYBR Green PCR Master Mix (Thermo Fisher Scientific).

**Isolation of ARMMs**. Culture medium atop nascent or transfected cells was collected and pre-cleared of cellular debris by two consecutive centrifugations (2000×$g$ for 10 min and 10,000×$g$ for 30 min). The medium was then passed through a 0.2-μm filter (Sarstedt) and subjected to ultracentrifugation using the SW41 Ti rotor in a L8-M centrifuge (Beckman) at 180,000×$g$ for 90 min. Pellets were washed twice, re-suspended in phosphate-buffered saline (PBS) and re-loaded onto a sucrose gradient of 10 different sucrose concentrations from top to bottom (0.2–2 M) and centrifuged at 180,000×$g$ for 18 h. Fractions were then carefully collected at 1 ml each from the bottom of the tube. All fractionated samples were diluted with PBS and subjected to centrifugation at 180,000×$g$ for 90 min to pellet the vesicles.

**Nanoparticle tracking analysis**. EVs or ARMMs were analyzed and quantitated by the NanoSight LM10 instrument with the Nanoparticle Tracking Analysis (NTA) software (Malvern). Samples containing vesicles were diluted with PBS and recorded for 60 s with 5 repeats. Videos were analyzed by the NTA software to obtain an average of the five repeats for each sample.

**Protein sequence analysis by LC-MS/MS**. LC-MS/MS and peptide analysis were done by the Taplin Biological Mass Spectrometry Facility at the Harvard Medical School. Briefly, gel slices from Coomassie blue-stained SDS-PAGE were cut into approximately 1 mm³ pieces, and subjected to trypsin digestion. Peptides were extracted and reconstituted in 5–10 μl of HPLC solvent A (2.5% acetonitrile, 0.1% formic acid). A nano-scale reverse-phase HPLC capillary column was created by packing 2.6 μm C18 spherical silica beads into a fused silica capillary (100 μm inner diameter × ~30 cm length). After equilibrating the column each sample was loaded via a Famos auto sampler (LC Packings, San Francisco, CA) onto the column. A gradient was formed and peptides were eluted with increasing concentrations of solvent B (97.5% acetonitrile, 0.1% formic acid). As peptides eluted they were

subjected to electrospray ionization and then entered into an LTQ Orbitrap Velos Pro ion-trap mass spectrometer (Thermo Fisher Scientific). Peptides were detected, isolated and fragmented to produce a tandem mass spectrum of specific fragment ions for each peptide. Peptide sequences (and hence protein identity) were determined by matching protein databases with the acquired fragmentation pattern by the software program, Sequest (Thermo Fisher Scientific). All databases include a reversed version of all the sequences and the data were filtered to between a 1 and 2% peptide false discovery rate.

**CRISPR knockout of *ARRDC1*.** The PX330 vector system (Addgene) was used to make CRISPR constructs. Two CRISPR constructs were designed to target exon 1 (guide sequence: GCACCGCTGCCGTTCCGAGG) and exon 6 (guide sequence: CGACGTACGGACCATTGCGG) of the human *ARRDC1* gene. The PX330-CRISPR-ARRDC1 constructs were transfected into the 293 T cells. Single stable cell clones were isolated and screened by genomic PCR for indels (insertions or deletions) in the ARRDC1 gene. Single clones with ARRDC1 knockout were further confirmed by ARRDC1 western blotting.

**Immunofluorescence staining and confocal imaging.** HEK293T cells grown on glass coverslips were incubated with isolated ARMMs for 2 h, washed with PBS three times and then fixed in 4% paraformaldehyde (Sigma) for 20 min on ice. Cells were permeabilized using 0.1% Triton X-100 in PBS for 5 min on ice and then blocked in the blocking buffer (2% bovine serum albumin in PBS) for 1 h at room temperature. The cells were incubated with primary antibody, followed by Alexa Fluor-labeled secondary antibodies (Invitrogen, 1:500). The glass coverslips were mounted on glass slides using Prolong Gold Antifade Mountant with DAPI (Thermo Fisher Scientific). Image acquisition was carried out using a Leica TCN-NT laser-scanning confocal microscope (Leica) equipped with air-cooled argon and krypton lasers. Images were processed using ImageJ.

**ARMMs transfer in transwell assay.** Transfected cells were washed thoroughly and seeded atop a 0.4-μm transwell membrane (Costar) for 24 h. The transwells were then transferred to a plate containing untransfected HEK293T cells. ARMMs transfer was allowed to proceed for 30 h before harvesting. Alternatively, purified ARMMs resuspended in PBS were added to culture medium containing untransfected HEK293T cells and were incubated for 24–30 h before harvesting.

**Statistical analyses.** All analyses of data between two groups were performed by unpaired Student's t-test. Data were presented as the mean ± standard deviation (s. d.). Differences were considered significant when $p < 0.05$.

**Data availability.** The proteomic data have been deposited into the PRIDE/ProteomeXchange database with identifier PXD006416. The data that support the findings of this study are available from the corresponding author upon request.

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

## Acknowledgements

The authors thank Ross Tomaino at the Taplin Mass Spectrometry Facility of the Harvard Medical School for his advice on the analysis of the proteomic data. This work was supported in part by a National Institutes of Health R01grant (R01 HL114769) and by funding from the Blavatnik Biomedical Accelerator Fund.

## Author contributions

Q.L. conceived the project. Q.W. and Q.L. designed the experiments, discussed the results, analyzed the data and wrote the paper. Q.W. performed all experiments.

## Additional information

**Competing interests:** The authors declare no competing financial interests.

