## [Peer Review File · Nature Communications]

Reviewers' comments:

Reviewer #1 (Remarks to the Author):

The manuscript by Wang et al. investigates the role of ARMM-type extracellular vesicles (EVs) in mediating NOTCH signaling at a distance. It starts with a discovery proteomics approach identifying proteins that are differentially enriched between EVs that have been purified from control or ARRDC1-overexpressing cells. At the top of the list was an E3 ubiquitin ligase, ITCH, that interacts with NOTCH, and NOTCH2 was also enriched. The followup experimentation shows that ARRDC1 interactions with ITCH can recruit NOTCH2 into EVs. ADAM10 was also enriched and is a proteinase that cleaves the extracellular domain of NOTCH2, as a first step before gamma-secretase releases the ICD. The authors show that ADAM10 does indeed promote maturation of NOTCH2 in ARMMs. They also show that tagged ARRDC1 and tagged NOTCH2 are transferred in a co-culture assay to recipient cells. Furthermore, this leads to activation of NOTCH transcription (Hes1 and Hes5 genes), which is dependent on ARRDC1 in the donor cell and gamma secretase in the recipient cells. This is overall a very interesting paper that demonstrates transfer of ARMM cargoes to recipient cells with a functional effect. As NOTCH signaling mediates cell fate and patterning, this long distance mechanism is very interesting and could change the way we think about morphogenic gradients both in development and in diseases such as cancer. Overall, it is well executed, with many controls and rescue experiments. The paper is also well written and was a pleasure to read. I believe it will be of interest to a wide readership and impact multiple fields, including EVs, Developmental Biology, Cell Biology, and anyone interested in cellular communication. I have a few relatively minor suggestions, as follows:

1. The Western blots do not have molecular marker indications. Please fix.
2. Also, there should be quantitations of the Western blot data from ≥ 3 independent replicates.
3. The n values used in statistics and the number of independent replicates should be detailed in the figure legends.
4. Fig 3C, the IP results are not very impressive and should be done in both directions to show specificity of the interaction.
5. Fig 5b, only the mature form of NOTCH2 is shown. Is the full length form of NOTCH2 now in ARMMs when ADAM10 is knocked down? What about in cell lysates – the full blots should be shown. Also, why is there no difference in mature NOTCH2 in cell lysates with ADAM10 knockdown?

Reviewer #2 (Remarks to the Author):

Wang and colleagues present a manuscript that used a proteomics strategy to initially describe cargo proteins present in ARMMs microvesicles, a specialized sub-type of extracellular vesicles. Since ARMMs MVs are similar in size to exosomes, but are formed by a different mode of biogenesis, a HEK293T cell line model with forced overexpression of ARRDC1-GFP was used. Since expression of this protein is sufficient to drive the budding and production of ARMMs MVs. Proteomics analyses identified over 100 cargo proteins in ARMM MVs. Comparison to EVs that were not over-expressing ARRDC1-GFP (but only GFP) and are hence not forced to produce ARMMs MVs were used for comparison.

Identification of NOTCH 1 and NOTCH2 was observed to be enriched in the ARMMs MVs and functionally investigated. The authors present an interesting story that suggests that NOTCH cargo in ARMMs MVs could participate in non-canonical NOTCH signalling, independent of cell-cell contact and possible at distant sites. Overall this is a straightforward manuscript, demonstrating a possible function of ARMMs EVs in intercellular signalling.

Comments

- 1) Very hard to evaluate the proteomics data, based on what is presented.
 - a. How many biological replicates were conducted?
 - b. The number of presented peptides; these are unique peptides or total spectra?
 - c. Supplemental tables need to be presented as proper Excel files.
 - d. Raw data needs to be deposited to a public depository.
 - e. Very low proteomic coverage compared to recent studies. Looking at the coomassie stained gels one would expect 100s-1000s of identified proteins? What is going on here? It's even a bit strange that nobody from the Taplin Biological Mass Spectrometry Facility is on this paper (or at least acknowledged)?

- 2) Since the proteomics experiments seem to be only semi-quantitative it is not clear how the comparison of general EVs vs. ARMMs MVs was actually controlled. Looking at figure 1d/e it seems there are much more vesicles in the ARRDC1-GFP condition as compared to the GFP control. The gel also shows a more intense protein staining. As a result of this, by definition more spectral counts will be obtained for most proteins in the ARRDC1-GFP condition, hence suggesting enrichment? Why wouldn't there be any proteins with enrichment in exosome-like vesicles?
 - a. Along these lines: The way the authors isolate ARMMs EVs is the same protocol used for isolating exosomes. Their final pellet is likely a mixed population of EVs and Exosomes (indeed they comment that exosomes markers, such as CD9, are expressed and enriched in that fraction). Why not isolate ARMMs EVs by Antibody capture? It's a membrane protein and they have a well working antibody?
 - b. ARRDC1 is also expressed in exosomes (according to ExoCarta) so their ARRDC1-positive fractions in the sucrose gradient might contain both kinds of vesicles.

- 3) Can these results be reproduced in additional cell line models, possibly without the forced over-expression of ARRDC1-GFP?

- 4) Many of the Western blots are hard to evaluate since they only show small sections of the gel without MW markers. It would be nice to see full gels and possibly coomassie stained membranes.

- 5) Figure 2b: Could be generated as a Protter figure for much better visualization (<http://wlab.ethz.ch/protter/start/>). At least as a supplement. That way we can all see the identified peptides and where they align.

- 6) Page 6; line 119: There are several proteomics datasets published that have reported on Notch proteins in exosome enriched EVs (PMIDs: 27894104, 24434149), just to name a few. In addition they are also reported in ExoCarta?

- 7) Fig 1C shows anti-GFP but not anti-ARRDC1.

- 8) RNAi experiments: Why not show knock-down with western blotting?

- 9) Transwell assay in Figure 4: It is somewhat hard to imagine that in 24 hours enough EVs could be transferred for such strong functional effects. It would be essential to support these data by imaging, showing that EVs are internalized. Dose and time-course dependent experiments should be conducted. In the second part of the experiment where ARMMs MVs are isolated and added to the recipient cells. How many MVs? Again time and dose-dependency should be shown. In addition, what would happen if these MVs are denatured, for example by boiling?

Overall, this is an interesting manuscript, providing proteomics data on ARMMs MV cargo. The identification of NOTCH proteins in these MVs was further evaluated, as a means of non-canonical

signalling. This reviewer felt that both for the proteomics studies, as well as many of the functional studies, additional controls are required to fully evaluate the quality of this manuscript.

We thank the reviewers for their constructive comments. Changes made to the manuscript are highlighted (vertical line to the left of changed text in a paragraph, or boxed for text). Below I detail our responses to each of the reviewers' critiques. For clarity, reviewer comments are italicized and followed by our specific responses.

Response to comments by reviewer #1

1. The Western blots do not have molecular marker indications. Please fix.

Response: The molecular weight markers are now added to all Western blots in the revised Figures.

2. Also, there should be quantitations of the Western blot data from ≥ 3 independent replicates.

Response: The quantitation of the Western blot data from 3 independent replicates was added for Fig. 3d. For other Western blots, the differences between controls and subjects (e.g. the amount of NOTCH2 in EV fractions from GFP vs. ARRDC1-GFP cells) were more pronounced and thus were not quantified. Nevertheless, all Western blotting data presented in the manuscript were confirmed by independent biological replicates.

3. The n values used in statistics and the number of independent replicates should be detailed in the figure legends.

Response: This information is included in the Figure legends as suggested.

4. Fig 3C, the IP results are not very impressive and should be done in both directions to show specificity of the interaction.

Response: We have used the Flag antibody to do the IP as the reviewer suggested. The interaction between Flag-ITCH and ARRDC1-HA is robust, and consistent with the previous HA-antibody IP result, the interaction is significantly reduced when the two PPXY motifs in ARRDC1 are deleted. The new Flag-IP data is included along with the previous HA-IP result in Fig. 3c.

5. Fig 5b, only the mature form of NOTCH2 is shown. Is the full length form of NOTCH2 now in ARMMs when ADAM10 is knocked down? What about in cell lysates – the full blots should be shown. Also, why is there no difference in mature NOTCH2 in cell lysates with ADAM10 knockdown?

Response: The full blots with both full length and short form NOTCH are now included in the revised Fig. 5c. Again the data showed the presence of the shorter NOTCH but not the full length NOTCH in ARMMs. We did not observe a decrease in mature (cleaved) NOTCH2 or an increase in the full length NOTCH2. This is likely due to the transient nature of ADAM10 knockdown by siRNA: much of the NOTCH2 receptor in the cells has already been cleaved into the shorter form before ADAM10 knockdown takes effect.

Unlike total cell lysates, EV/ARMMs were collected during the peak siRNA knockdown period (36-72 hours), allowing us to assess the effect of ADMA10 siRNA on NOTCH2 secretion into ARMMs.

Response to comments by reviewer #2

1. Very hard to evaluate the proteomics data, based on what is presented.

a. How many biological replicates were conducted?

Response: For the proteomics experiment, we harvested the vesicles from 4 different batches of production. The vesicle yield for each batch was relatively low for proteomic study, so we combined those samples together for the proteomics study. For this paper we used the proteomics as a discovery tool. Many of the findings, including enrichment of ARRDC1, NOTCH2, and ITCH in ARMMs, were later confirmed by Western blotting.

b. The number of presented peptides; these are unique peptides or total spectra?

Response: Total peptide number for each protein was used in Table 1 and in the Supplementary Tables (1-4). Positions of unique peptides detected by mass-spec for NOTCH2 are shown in Fig. 2a, with sequences included in the Supplementary Fig.S1.

c. Supplemental tables need to be presented as proper Excel files.

Response: Supplementary mass-spec results are included as Excel files as suggested. All other Supplementary information is included in a single Word file as required by *Nature Communications*.

d. Raw data needs to be deposited to a public depository.

Response: Thank you for the suggestion. The proteomic data that support the findings of this study have been deposited into the PRIDE/ProteomeXchange database with identifier PXD006416.

e. Very low proteomic coverage compared to recent studies. Looking at the coomassie stained gels one would expect 100s-1000s of identified proteins? What is going on here? It's even a bit strange that nobody from the Taplin Biological Mass Spectrometry Facility is on this paper (or at least acknowledged)?

Response: We apologize for the confusion here. We actually identified much more proteins. In total, we identified over 1000 proteins in both GFP and ARRDC1-GFP samples. Majority of the proteins didn't show significant change between GFP and ARRDC1-GFP samples. We used '> 1.5 fold' enrichment in ARRDC1-GFP samples as the cut-off. Therefore, we only included the proteins that increased more than 1.5 fold in Supplementary Table 1. Mass-spectrometry and raw data analysis were done through a fee-based service by Ross Tomaino at the Taplin Mass Spectrometry Facility of the Harvard Medical School. This is now indicated in the Acknowledgements.

2) *Since the proteomics experiments seem to be only semi-quantitative it is not clear how the comparison of general EVs vs. ARMMs MVs was actually controlled. Looking at figure 1d/e it seems there are much more vesicles in the ARRDC1-GFP condition as compared to the GFP control. The gel also shows a more intense protein staining. As a result of this, by definition more spectral counts will be obtained for most proteins in the ARRDC1-GFP condition, hence suggesting enrichment? Why wouldn't there be any proteins with enrichment in exosome-like vesicles?*

Response: We have shown previously that ARMMs are non-exosomal and do not contain significant amount of classic exosomal proteins such as CD9 and CD81 (Nabhan 2012, PNAS). Consistent with this, CD9 and CD81 were found to be unchanged between general EVs and ARMMs EVs. So we used these exosomal markers to control the comparison. Yes there are more proteins in EVs from ARRDC1 overexpressing cells, as we would expect from more vesicle production (general EVs plus ARMMs).

a. Along these lines: The way the authors isolate ARMMs EVs is the same protocol used for isolating exosomes. Their final pellet is likely a mixed population of EVs and Exosomes (indeed they comment that exosomes markers, such as CD9, are expressed and enriched in that fraction). Why not isolate ARMMs EVs by Antibody capture? It's a membrane protein and they have a well working antibody?

Response: Yes, our isolated EVs are a mixture of exosome and ARMMs. It is difficult to separate those two kinds of EVs based on standard density-based separation methods as they are of similar size (as shown in Fig. 1b). It would be great to isolate ARMMs via specific surface markers. However, prior to our current study, we did not know what plasma membrane proteins are specific for ARMMs. Therefore, it was not possible to use antibody to capture ARMMs. Now that this study has identified membrane proteins potentially unique in ARMMs, we are trying to isolate the ARMMs based on these markers. However we found it very challenging to scale up the method to isolate enough vesicles for mass-spec analysis.

b. ARRDC1 is also expressed in exosomes (according to ExoCarta) so their ARRDC1-positive fractions in the sucrose gradient might contain both kinds of vesicles.

Response: See response above. Also ExoCarta in general does not distinguish ARMMs and exosomes. Since there is no good method to separate exosomes from ARMMs, and *vice versa*, we believe that the ExoCarta database includes both kinds of vesicles (and likely other uncharacterized EVs)

3) *Can these results be reproduced in additional cell line models, possibly without the forced over-expression of ARRDC1-GFP?*

Response: Yes, thank you for this great suggestion. Now we include new data in Supplementary Fig. S2. We showed the presence of NOTCH2 shorter form in EVs from three additional cell lines (HCC1419, A549 and MCF-7) (Fig. S2a). More importantly,

we showed that knockdown of ARRDC1 reduces NOTCH2 present in EVs in MCF-7 cells (Fig. S2b). These new data support our main findings and indicate that NOTCH incorporation in ARMMs is generalizable in diverse cell types. The knockdown data in MCF-7 cells (Supplementary Fig. 2b) as well as the knockout data in HEK293 cells (Fig. 2c) were both with endogenous ARRDC1 (i.e. without ARRDC1 overexpression).

4) *Many of the Western blots are hard to evaluate since they only show small sections of the gel without MW markers. It would be nice to see full gels and possibly coomassie stained membranes.*

Response: Molecular weight markers are now added for all Western blots. Full gels from which the sections were cropped are included as Supplementary information.

5) *Figure 2b: Could be generated as a Protter figure for much better visualization (<http://wlab.ethz.ch/protter/start/>). At least as a supplement. That way we can all see the identified peptides and where they align.*

Response: NOTCH2 protein is relatively big (2,471 amino acids) to be visualized by Protter. However, we added the detailed sequence information of the peptides in the Supplementary Fig. 1.

6) *Page 6; line 119: There are several proteomics datasets published that have reported on Notch proteins in exosome enriched EVs (PMIDs: 27894104, 24434149), just to name a few. In addition they are also reported in ExoCarta?*

Response: When we first obtained the mass-spectrometry dataset, NOTCH2 was not included in the ExoCarta database. The database has since been updated to include NOTCH2. In light of this and the two references cited by the reviewer, we removed the previous statement. However, since ExoCarta as well as the studies that contributed to database does not distinguish among the different forms of EVs, it is very likely that NOTCH2 detected in EVs was coming from ARMMs.

7) *Fig 1C shows anti-GFP but not anti-ARRDC1.*

Response: Because ARRDC1 was fused to GFP, we used anti-GFP antibody to detect both GFP and ARRDC1-GFP. This allows for a better comparison between control GFP and ARRDC1-GFP samples. If we use anti-ARRDC1 antibody, we would only detect ARRDC1 but not the control GFP. We modified the label for Fig 1C to avoid the confusion.

8) *RNAi experiments: Why not show knock-down with western blotting?*

Response: For Fig. 2d, we now include ITCH Western blot result that is consistent with the mRNA decrease by ITCH siRNA transfection. For ADAM10, we tried 3 different commercial antibodies, but could not find a suitable one for Western blot. Thus we used qRT-PCR to show the knockdown of the ADAM10 mRNA.

9) Transwell assay in Figure 4: It is somewhat hard to imaging that in 24 hours enough EVs could be transferred for such strong functional effects. It would be essential to support these data by imaging, showing that EVs are internalized. Dose and time-course dependent experiments should be conducted. In the second part of the experiment where ARMMs MVs are isolated and added to the recipient cells. How many MVs? Again time and dose-dependency should be shown. In addition, what would happen if these MVs are denatured, for example by boiling?

Response: Many studies have shown that EVs are taken up by recipient cells very quickly, sometimes within 30 min (references # 32 and 33: PMID:20136776, PMID:15284116; also reviewed in PMC4122821). We have performed confocal imaging to show that the uptake of ARMMs by the recipient HEK293T cells also occurs fairly quickly--within 2 hours of incubation (Supplementary Fig. S3). The EVs number we used in the non-transwell experiments (where ARMMs were isolated and added) was about 2×10^9 per well (in 6-well plate). Given that NOTCH receptor proteins need to be active to induce gene expression, we anticipate that denaturing the ARMMs would abolish the functional transfer of NOTCH to recipient cells.

REVIEWERS' COMMENTS:

Reviewer #1 (Remarks to the Author):

My comments are mostly addressed. However, the quantitation that is supposed to be in Fig 3d does not seem to be in there. The only quantitation that I see is of mRNA levels for ITCH after ITCH KD. The other sub panels in Fig 3d are Western blots but do not show quantitations from n=3 at least that I can tell unless the numbers under the blot are from n=3. If that is the case, it would be better represented as a bar graph, as typically numbers under a lane are from just those lanes. The blots are also not labeled as to whether they are from cell lysates or EVs and do not seem to be adequately described in the Figure legend.

In addition, the changes in the Supplement were not something I asked for, but I notice that the panels in Supp Figs 2 and 3 are empty -- I think they are place holders and the real data were not put in.

So, some pieces of the manuscript do not seem to have been fully updated before resubmission.

Reviewer #2 (Remarks to the Author):

Considering the revised manuscript and all supplemental data and the answers to my previous question, I believe the authors have addressed all my concerns.

This is an interesting story/mechanism and should hence be published in Nature Communications .

We thank the reviewers for their constructive comments. Below I detail our responses to each of the reviewers' critiques. For clarity, reviewer comments are italicized and followed by our specific responses.

Response to comments by reviewer #1

1. *My comments are mostly addressed. However, the quantitation that is supposed to be in Fig 3d does not seem to be in there. The only quantitation that I see is of mRNA levels for ITCH after ITCH KD. The other sub panels in Fig 3d are Western blots but do not show quantitations from n=3 at least that I can tell unless the numbers under the blot are from n=3. If that is the case, it would be better represented as a bar graph, as typically numbers under a lane are from just those lanes. The blots are also not labeled as to whether they are from cell lysates or EVs and do not seem to be adequately described in the Figure legend.*

Response: We have added the quantitation to Fig. 3d. The data were from 3 independent experiments and are now presented in a bar graph as suggested. The blot in the left panel is from cell lysates, the right blots are from EVs. They are now clearly labeled.

2. *In addition, the changes in the Supplement were not something I asked for, but I notice that the panels in Supp Figs 2 and 3 are empty -- I think they are place holders and the real data were not put in. So, some pieces of the manuscript do not seem to have been fully updated before resubmission.*

Response: During initial PDF conversion, some Western blot images in Supp Figs were inadvertently corrupted / deleted. We realized the error and had supplied a correct version, which for some reason was not seen by the reviewer. We apologize for this. Now the correct PDF conversion along with the original data file is provided.

Response to comments by reviewer #2

Considering the revised manuscript and all supplemental data and the answers to my previous question, I believe the authors have addressed all my concerns. This is an interesting story/mechanism and should hence be published in Nature Communications.

Response: Thank you.